# Wheat Protein Hydrolysates Improving the Stability of Purple Sweet Potato Anthocyanins under Neutral pH after Commercial Sterilization at 121 °C

**DOI:** 10.3390/foods13060843

**Published:** 2024-03-10

**Authors:** Yaping Feng, Bingqian Qiao, Xue Lu, Jianhui Xiao, Lili Yu, Liya Niu

**Affiliations:** School of Food Science and Engineering, Jiangxi Agricultural University, 1101 Zhimin Road, Nanchang 330045, China; fyp981026@163.com (Y.F.); 15797716194@163.com (B.Q.);

**Keywords:** wheat protein hydrolysates, purple sweet potato anthocyanins, commercial sterilization, heat stability, complex formation, pH

## Abstract

Anthocyanins are prone to degradation and color fading after sterilization. This work examined the potential of wheat protein hydrolysates (WPHs, 40 g/L) in improving the stability of purple sweet potato anthocyanins (PSPAs) under a pH of 6.8 after sterilization at 121 °C followed by storage. Results showed that WPHs increased the thermal degradation half-life of PSPAs 1.65 times after sterilization. Compared to PSPAs alone, after being stored at 37 °C and 45 °C for 7 days, the retention concentration of PSPAs with WPHs was 5.4 and 32.2 times higher, and the color change of PSPAs with WPHs decreased from 6.19 and 10.46 to 0.29 and 0.77, respectively. AFM data, fluorescence and UV spectrograms indicated the formation of complexes between PSPAs and WPHs by hydrophobic attraction confirmed by zeta-potential data. PSPAs with WPHs had stable particle size and zeta potential, which may also significantly increase the concentrations after digestion and antioxidant power of PSPAs. This work indicated that the assembled PSPAs composite structure by WPHs significantly reduced the degradation of PSPAs at a pH of 6.8 after sterilization at 121 °C followed by long-term storage.

## 1. Introduction

Purple sweet potato anthocyanins (PSPAs) derived from purple sweet potato, which are mainly composed of cyanidin and peonidin, are widely used as natural colorants in foods [1]. The main chemical structure of PSPAs is anthocyanin and peony glycosides in the form of monoacetylation and deacetylation, and the proportion of acetylated anthocyanins are more than 93% with a variety of desirable functional attributes to human health, including antioxidant properties, anticarcinogenic, visual acuity and dermal health [2,3,4,5]. Therefore, the addition of PSPAs is an effective way to improve the antioxidant properties and bioactivity of food, including protein-based food, such as milk, soy milk, coconut milk and peanut milk with a neutral pH. However, in neutral pHs ranging from 6.0 to 7.4 after commercial sterilization at 121 °C, the structure of anthocyanins changed from a flavylium cation to chalcone and a quinone group, which was more unstable. The low stability seriously limits the application of anthocyanins [6,7].

In addition, food processing with the conditions of heat, light and oxygen makes PSPAs prone to degradation. Thus, many works have reported that proteins, carbohydrates and amid acids can improve the stability of anthocyanins via approaches including encapsulation, self-assembly, co-pigmentation and binding with metal ions [8,9]. In neutral pH, the nucleophilic amino acids on the side chain of protein can covalently bind to the quinone formed by anthocyanins to enhance the stability of anthocyanins [10,11].

Compared to the protein, wheat protein hydrolysates (WPHs) obtained from wheat proteins by enzymatic hydrolysis gifted the characteristics, including good water solubility, stable dispersion, easy absorption and strong biological activity [12,13,14]. Moreover, the content of several special amino acids (such as glutamine and proline) in WPHs with a variety of nutritional regulatory functions in the process of cell division, differentiation, protein synthesis and interconversion of various substance metabolism is much higher than that of any other plant protein hydrolysates [15,16]. Also, it has been proven that WPHs can improve the stability of chlorogenic acid during simulated gastrointestinal digestion, promote the efficiency of chlorogenic acid entering intestinal epithelial cells and increase its intracellular antioxidant activity [17]. Therefore, the interaction between WPHs with both functionality and clean labels and PSPAs may be considered to improve the stability of PSPAs. Currently, studies on the stability of PSPAs at neutral pHs after commercial sterilization at 121 °C when they are co-dissolved with WPHs are lacking.

The aim of this paper was to evaluate the potential for improving the stability and maintaining antioxidant properties of PSPAs by WPHs stored at 37 °C and 45 °C for 7 days after commercial sterilization at 121 °C in neutral pH. The degradation kinetics, storage stability, antioxidant activity and in vitro digestion as well as the interaction of PSPAs with WPHs were also investigated. 

## 2. Materials and Methods

### 2.1. Materials

Purple sweet potato anthocyanins (PSPAs) were purchased from Xi’an Aomeng Sisheng Biotechnology Co., Ltd. (Xi’an, China). Wheat protein hydrolysates (WPHs) were purchased from Zhongshi Duqing Biotechnology Co., Ltd. (Heze, China). Sodium citrate and citric acid were purchased from Weifang Ensign Industry Co., Ltd. (Weifang, China). Sodium hydroxide (food grade) was purchased from Binhua Group Co., Ltd. (Binhua, China). Pepsin from porcine gastric mucosa (3200–4500 U/mg protein) and trypsin from porcine pancreas were purchased from Sigma Co., Ltd. (Livonia, MI, USA). All other chemicals used in this work were of analytical grade and purchased from Sinopharm Co., Ltd. (Shanghai, China).

### 2.2. Preparation of Samples

A mixed solution of 40 g/L of WPHs and 4 g/L of PSPAs was dissolved in the buffer solution of a pH of 6.8, then commercially sterilized (121 °C, 0.1 MPa, 20 min) and stored for 7 days at 37 °C and 45 °C in constant temperature incubators for subsequent testing [18].

### 2.3. Measurement of Degradation

#### 2.3.1. Color

The color changes of the samples were monitored by a colorimeter (Color Quest XE, HunterLab, Reston, VA, USA) according to the previous methods [19]. Color parameters including L* (lightness), a* (red to green) and b* (yellow to blue) were recorded. Additionally, the total color difference (ΔE) values, the chroma (C*) and hue angle (h) were calculated.
(1)$$\Delta E=\sqrt[{2}]{{\Delta L}^{*2}+{\Delta a}^{*2}+{\Delta b}^{*2}}$$
(2)$$C^{*}=\sqrt[{2}]{{\Delta a}^{*2}+{\Delta b}^{*2}}$$
(3)$$h=\tan^{-1}(\frac{b^{*}}{a^{*}})$$

#### 2.3.2. Anthocyanin Content

The anthocyanin content in the samples was determined by a visible spectrophotometer (V-5600 spectrophotometer, Shanghai, China) [20]. Briefly, each sample was diluted from 0.5 mL to 5 mL by a potassium chloride buffer (0.025 mol/L, pH 1.0) and sodium acetate buffer (0.4 mol/L, pH 4.5) and then incubated at 23 °C for 30 min. The absorbance values at the maximum absorption peak (A_527_) and 700 nm (A_700_) were measured. The anthocyanin content was calculated according to the following equation:(4)$$H=\frac{\Delta A\times \mathrm{MW}\times \mathrm{DF}\times 10^{3}}{\varepsilon \times 1}$$
where H is the anthocyanin content (mg/L), ΔA is (A_527_ − A_700_) _pH 1.0_ − (A_527_ − A_700_) _pH 4.5_, MW is the relative molecular weight at 449.2 based on cyanidin-3-O-glucoside, DF is the dilution time, 1 is the range of light, ε is the molar extinction coefficient which was calculated to 26,900 L/(mol*cm) for cyanidin-3-O-glucoside.

#### 2.3.3. Brown Indexes (BI) and Polymeric Color Indexes (PCI)

BI and PCI were determined by the metabisulfite bleaching method [1,21]. Samples were diluted in a citrate-phosphate solution (1:20 *v*/*v*, pH 2.2). An amount of 1 mL of metabisulfite (1 mol/L) or citrate-phosphate buffer was added to 4 mL of diluted sample. After equilibrium for 20 min, the absorbance values at 420 nm, 527 nm and 700 nm were measured. PCI was determined by Equations (5)–(7). BI was calculated by using the following equation:(5)$$\mathrm{PCI}=\frac{\mathrm{Polymeric}\;\mathrm{color}}{\mathrm{Color}\;\mathrm{density}}$$
(6)$$\mathrm{Polymeric}\;\mathrm{color}=\left[\left(A_{420}-A_{270}\right)+\left(A_{527}-A_{700}\right)\right]\times \mathrm{dilution}\;\mathrm{factor}$$
where A is the absorbance of the samples mixed with metabisulfite at a specific wavelength.
(7)$$\mathrm{Color}\;\mathrm{density}=\left[\left(A_{420}-A_{700}\right)+\left(A_{527}-A_{700}\right)\right]\times \mathrm{dilution}\;\mathrm{factor}$$
(8)$$\mathrm{BI}=\frac{A_{420}}{A_{527}}$$
where A is the absorbance of the samples mixed with citrate-phosphate buffer at a specific wavelength.

### 2.4. Degradation Kinetics of Anthocyanins

The solutions were subjected to sterilization tests at 80 °C and 121 °C as well as a light test at 25 °C, and the measured anthocyanin concentrations were fitted by Equation (9) for the first-order degradation kinetics. The half-life (t_1/2_) was calculated according to Equation (10).
(9)$$\frac{c_{t}}{c_{0}}=e^{-kt}$$
(10)$$t_{\frac{1}{2}}=-\ln\left(0.5\right)\cdot k^{-1}$$
where c_0_ is anthocyanin content before sterilization, c_t_ is anthocyanin content after sterilization, *t* is time of sterilization, *k* is the degradation rate constant.

### 2.5. Atomic force Microscope (AFM)

A total of 5 μL of the diluted solution was incubated on a split mica surface, and dried overnight at 23 °C [22,23]. Morphology images were collected by AFM Multimode VIII microscope (Bruker, Germany) with a nano-probe cantilever tip in the frequency range of 50~100 kHz in tapping mode.

### 2.6. Zeta Potential and Particle Size

The zeta potential and particle size of samples with a viscosity of about 12.5 mPa.s were determined by a Nanoscale-laser particle size analyzer (NanoBrook Omni, Brookhaven, MS, USA) at 25 °C in the PALS mode with the sample count rate of 250 kcps [24].

### 2.7. Spectral Characteristics

#### 2.7.1. UV–Vis Spectroscopy

The spectrophotometer (UV-5200PC spectrophotometer, Shanghai, China) was used to measure the absorption spectrum of samples in the wavelength range of 200–800 nm [19]. All measurement was carried out at 23 °C using a quartz cuvette of 1 cm.

#### 2.7.2. Fluorescence Spectroscopy

The fluorescence spectra of the samples were measured using a fluorescence spectrophotometer (970CRT, Shanghai, China) at an excitation wavelength of 280 nm and an emission wavelength of 250–450 nm. Δλ was set at 15 nm and 60 nm. By scanning at 260–320 nm and 260–450 nm wavelengths, the synchronized fluorescence spectrum of the sample was obtained.

### 2.8. Antioxidant Capacity

#### 2.8.1. DPPH Free Radical Scavenging Activity

The method was slightly modified according to Reza Safari [25]. The mixture of 1.0 mL DPPH solution with a concentration of 50.0 μg/mL and 0.1 mL samples was kept in the dark for 30 min. Deionized water was used as a blank control. The absorbance of the solution at a wavelength of 517 nm was used to calculate the activity of scavenging DPPH radicals as follows:(11)$$\eta_{\mathrm{DPPH}}\left(\%\right)=\left(1-\frac{A_{t}-A_{s}}{A_{c}}\right)\times 100$$
where A_c_ is the absorbance value of the sample solvent after reaction with DPPH solution, A_t_ is the absorbance value of the sample after reaction with DPPH solution, A_s_ is the absorbance value of the sample mixed with deionized water.

#### 2.8.2. ABTS Free Radical Scavenging Activity

The method was slightly modified according to Li Zhou [26]. A 25 μL sample was mixed with 3.0 mL working liquid of ABTS at 25 °C and kept for 10 min. Deionized water was used as a blank control. Then, the absorbance of the solution at a wavelength of 734 nm was used to calculate the activity of scavenging ABTS radicals as follows:(12)$$\eta_{\mathrm{ABTS}}\left(\%\right)=\left(1-\frac{A_{t}-A_{s}}{A_{c}}\right)\times 100$$
where A_c_ is the absorbance value of the sample solvent after reaction with ABTS solution, A_t_ is the absorbance value of the sample after reaction with ABTS solution, A_s_ is the absorbance value of the sample mixed with deionized water.

#### 2.8.3. Ferric Reducing Antioxidant Power (FRAP)

The method was slightly modified according to Muhammad Siddiq [27]. The samples and working solution of FRAP were added to the 96-well enzyme plate. After reaction at 37 °C for 10 min, the absorbance at 593 nm was measured with enzyme mark instrument (M2, Molecular Devices, Solana Beach, CA, USA). The standard curve was established with FeSO_4_·7H_2_O (y = 0.2247x + 0.1355), and the FRAP activity was expressed with mM Fe^2+^/g.

### 2.9. In Vitro Digestion

According to the method of M. Minekus [28], the in vitro digestion was performed in a simulated gastrointestinal environment. The simulative digestion was performed in gastric digestion for 2 h and intestinal digestion for 2 h. After each digestion step, the samples were collected and stored in the refrigerator at −80 °C for the subsequent measurement of anthocyanin concentration.

### 2.10. Statistical Analysis

The results obtained from three replicates were statistically analyzed using one-way analysis of variance (ANOVA) followed by Duncan’s multiple-range test (*p* < 0.05). IBM SPSS Statistics 25 was used for analyzing the data, and Origin 2021 was used for the pictures.

## 3. Results and Discussion

### 3.1. Color Stability and Anthocyanins Concentration

As can be seen from Figure 1 and Table 1, the color stability of PSPAs solutions after commercial sterilization at 121 °C for 20 min followed by storage at 37 °C or 45 °C for 7 days were significantly improved by the addition of WPHs. After the solutions with WPHs were stored at 37 °C or 45 °C for 7 days, the L* value, representing the brightness of the solution color, and the a* value, representing the reddish-green color, the b* value, representing the yellowish-blue color, and the chroma (C*) and hue angle (h) value, all exhibit smaller differences. Additionally, the ΔE of PSPAs with WPHs was less than 1.0, which was significantly lower than that of PSPAs alone, which means a better color stability of the solution. This indicated that the interaction between WPHs and PSPAs at a pH of 6.8 significantly improved the color stability of PSPAs during storage. The color of anthocyanin solutions was positively correlated with the anthocyanin content in the solution [10,11].

Figure 2 showed that the content of anthocyanins was kept in the PSPA solutions after different treatments. After sterilization, the anthocyanin content of PSPAs with WPHs was 0.27 times higher than that of PSPAs alone. However, after storage at 37 °C or 45 °C for 7 days, the retained anthocyanin content of PSPAs with WPHs was 5.4 times and 32.2 times higher than that of PSPAs alone. The structure of anthocyanins changed from the more stable flavylium cation to unstable chalcone and quinone base structures at a pH of 6.8, which was more susceptible to degradation under unfavorable conditions such as high temperature and light exposure [29]. Related studies also showed that the covalent binding between anthocyanins and amino acids was mainly caused by the nucleophilic amino acid and quinone base of anthocyanins, so the neutral environment at a pH of 6.8 promoted the covalent binding between PSPAs and WPHs [10,11].

### 3.2. BI and PCI

Figure 3 showed that the BI of PSPAs increased significantly with sterilization and storage treatments, and the BI exceeded 1 after storage at 37 °C and 45 °C for 7 days and increased with the increased storage temperature. After the addition of WPHs, the BI was less than 1, which was significantly lower than that of PSPAs alone. When the BI was greater than 1, it indicated that more brown substances were produced in the samples [30]. The brown substances were probably mainly generated by the degradation of PSPAs alone. In contrast, the brown substance in PSPAs with WPHs might be mainly generated by the degradation of anthocyanins and the Maillard reaction, but less brown substance was generated due to the protective effect of WPHs on PSPAs. In the process of thermal processing and storage, the anthocyanins were prone to degradation and polymerization for producing brown substances. For example, 2,4,6-trihydroxy benzaldehyde from anthocyanins decomposition was a brown substance and the process of the Maillard reaction also produced brown substances [31].

The PCI is also an important indicator to evaluate the degradation, polymerization and brown substance formation of anthocyanins. And the degradation of anthocyanins was accompanied by an increase in the PCI [31]. According to Figure 4, the PSPAs had higher PCI after storage for 7 days at 37 °C and 45 °C, and the PCI of the PSPAs gradually increased with an increase in storage temperature, which indicated that the increase in temperature accelerated the degradation of anthocyanins. The PCI of the anthocyanin solutions was significantly lower than that of PSPAs after the addition of WPHs.

### 3.3. Degradation Kinetics

The degradation kinetics of PSPAs with/without WPHs at 80 °C, 121 °C and 25 °C with light were shown in Figure 5. The linear relationship indicated that the degradation of PSPAs followed the first-order reaction kinetics, which is consistent with previous studies on thermal degradation of anthocyanins [32]. The degradation rate constant (*k*) and half-life of degradation were estimated according to Equations (6) and (7), listed in Table 2. The degradation of PSPAs was more significant at higher temperatures, and the degradation half-life of PSPAs was significantly prolonged by WPHs at each temperature (*p* < 0.05). The addition of WPHs reduced the *k* value of PSPAs by 48.9% and 39.4% at 80 °C and 121 °C, respectively. Correspondingly, the degradation half-life of PSPAs at 80 °C and 121 °C was prolonged by 0.96 and 0.65 times, respectively. Furthermore, the addition of WPHs greatly improved the photostability of PSPAs under the light condition of 25 °C, which reduced the *k* value by 74% and extended the half-life from 6.7 days to 26.3 days, which was 2.9 times longer. The PSPAs with WPHs still maintained a bright purple color after 12 days of storage at 25 °C with light.

Anthocyanins have four main chemical structures in solution, two of which are colorless: the methanol pseudobasic and chalcone, and the other two are colored: the blue-violet quinone base, and the red flavylium form. Their quantity determines the color of the solution, and they are transformable with each other in dynamic equilibrium [33]. The previous study has shown that anthocyanins may exist mainly as red flavonoids at a pH of 1~2, whereas at a pH of 3~5, flavonoid cations hydrate to form a carotenoid pseudobasic base, and at a pH of 6~10, quinone anions are gradually formed [34]. The degradation process of anthocyanins involves nucleophilic attack of C-2 by water to form the methanol pseudobasic. The C ring then opens to form the chalcone, which is further degraded to the brown product [35]. In the present study, WPHs showed significant potential to protect PSPAs from degradation. Reports have shown that the hydroxyl group of anthocyanins can interact with the amide carbonyl group on the peptide chain through hydrogen bonding and be stabilized by proline residues and hydrophobic interactions [36]. In our study, WPHs interacted with PSPAs to form a complex that inhibited the thermal degradation of anthocyanins, which was also confirmed by the results of AFM.

### 3.4. Changes of Particle Size and Morphology

AFM was used to visualize the aggregation complexes of PSPAs alone and PSPAs with WPHs after accelerated storage (Figure 6). Before storage, the particle size of PSPAs alone was smaller than that of PSPAs with WPHs (Table 3). After storage, the particle size of PSPAs alone or PSPAs with WPHs increased significantly. The results showed that accelerated storage resulted in the condensation of anthocyanins solution for increasing particles. The phenomenon of aggregation of anthocyanins in solution during storage was also found in the studies of others [37].

### 3.5. Zeta Potential

Zeta potential is a measure of the surface charge density of proteins and reflects the stability of protein solutions [24]. The greater the absolute value is, the greater the electrostatic repulsion force is [38]. It can be seen from Table 3 that the PSPAs alone showed the maximum value of zeta potential. The value of zeta potential of PSPAs with WPHs was lower than that of PSPAs. The relevant literature [38,39,40] pointed out that the interaction between polyphenols and proteins would raise the large particles in the solution and reduce the stability of the solution, and the decrease in zeta potential value of PSPAs with WPHs indicated the formation of a complex between them. However, during accelerated storage, the zeta potential values of the anthocyanin solution did not change significantly, indicating that the anthocyanin solution was stable.

### 3.6. UV–Vis Spectroscopy

UV–vis absorption measurement is a simple method and is applicable for the changes in hydrophobicity and the formation of complexes. The π→π* transition in the peptide bond of the protein results in UV absorption in the range from 180 nm to 230 nm. Aromatic side chains of tryptophan (Trp), tyrosine (Tyr) and phenylalanine (Phe) residues dominate absorption in the range from 230 nm to 300 nm [41]. With the increase in PSPA concentration, the absorption intensity of the UV spectrum at 527 nm gradually increased, and the peak position shifted from 273 nm to 284 nm (Figure 7). This result indicated that the addition of PSPAs changed the structure of WPHs to a more hydrophobic microenvironment of the aromatic amino acid residues in WPH nanoparticles [42,43].

### 3.7. Fluorescence Spectroscopy

When small molecular substances are added to the protein solution to form complexes, the fluorescence intensity decreases; this is called fluorescence quenching [44].

The intrinsic fluorescence of proteins responds to protein denaturation, conformational transitions and subunit binding, primarily by tryptophan and tyrosine residues at excitation wavelengths of 280 nm [45].

Figure 8 shows that the addition of PSPAs to WPH solutions resulted in the changes in the WPH fluorescence spectra derived from tryptophan and tyrosine [46], and the fluorescence intensity of WPHs gradually decreased with the increase in PSPA content in the solution. In addition, the λ_max_ of WPHs showed a red shift (359–373 nm) as the concentration of PSPAs increased, suggesting an increase in the polarity of the microenvironment near the tryptophan and tyrosine residues in WPHs due to the interaction of proteins with anthocyanins with the addition of PSPAs, and these results were similar to the related studies [6,43,45].

### 3.8. Synchronous Fluorescence Spectroscopy

Synchronous fluorescence spectroscopy is a common method to study protein conformation [47]. Synchronous fluorescence spectroscopy was obtained by simultaneous scanning at the excitation (λ_ex_) and emission (λ_em_) wavelengths with a constant wavelength interval (Δλ) between them [6]. These changes can be observed by selecting a specific wavelength interval of synchronous fluorescence spectroscopy. When the scanning interval is Δλ = 15 nm and Δλ = 60 nm, the fluorescence emission peaks are characteristic absorption of tyrosine residues and tryptophan residues [8]. Changes in the polarity of the surroundings affected the shift of the emission wavelength maximum (λ_max_) [47].

Figure 9a,b shows the effect of PSPAs on the synchronized fluorescence spectra of WPHs at Δλ = 15 nm and Δλ = 60 nm. According to Figure 9a, the synchronous fluorescence intensities at Δλ = 15 nm showed a significant decrease and a blue shift (298.7–296.7 nm) with the increase in concentration of PSPAs. A blue shift in the maximum emission wavelength indicated increased hydrophobicity and decreased polarity [8]. As can be seen from Figure 9b, the synchronous fluorescence intensities at Δλ = 60 nm showed a significant decrease and a slight red shift (349.1–351.1 nm) with the increase in PSPA concentrations. The red shift in the maximum emission wavelength indicated decreased hydrophobicity and increased polarity, indicating that the amino acid residues were gradually exposed [48].

### 3.9. Antioxidant Ability

The DPPH and ABTS radical scavenging capacity and FRAP of WPHs, PSPAs alone and PSPAs with WPHs are shown in Figure 10. It could be seen from the figure that the antioxidant capacity of PSPAs with WPHs was significantly higher than that of PSPAs or WPHs alone. As a natural pigment, PSPAs gifted a variety of biological activities, such as anti-cancer and antioxidation activity [7]. The bioactivity of wheat protein was also enhanced by the enzymatic hydrolysis of wheat protein into WPHs [49], indicating that the interaction between the WPHs and PSPAs improved the antioxidant capacity of the solution.

From Figure 10a, the DPPH free radical scavenging ability of PSPAs with WPHs decreased by 2.9% and 5.6% after storage for 7 days at 37 °C and 45 °C; this was significantly lower than 11% and 24% of PSPAs alone. It can be seen from Figure 10b that the ABTS free radical scavenging capacity of PSPAs with WPHs, PSPAs and WPHs decreased significantly after accelerated storage, indicating that accelerated storage caused certain damage to the biological activities of WPHs and PSPAs. But PSPAs with WPHs decreased by 3% and 5% after storage for 7 days at 37 °C and 45 °C which was significantly lower than 28% of PSPAs. It can be seen from Figure 10c that the total antioxidant capacity of PSPAs with WPHs did not decrease significantly after storage, and it was more stable than PSPAs alone.

### 3.10. In Vitro Digestion

The stability of PSPAs in the intestine was explored by in vitro simulated digestion [9] that was shown in Figure 11. The results revealed a very low digestive retention of PSPAs. However, the retention rate of PSPAs with WPHs after digestion was 1.9 times that of PSPAs alone. During gastric digestion (0–120 min), PSPAs existed in the structure of yellow terminal salt cation with high stability due to the low acidity of the gastric environment. During the period of 60–120 min, the PSPA content in the solution increased temporarily and then decreased. This might be the fact that the acidic environment changed the structure of PSPAs from methanol pseudobase and chalcone to yellow closing salt cation structure for increasing the content of PSPAs, but gastric digestion reduced the content of PSPAs to a certain extent [23]. During intestinal digestion (120–240 min), the degradation rate of PSPAs was significantly faster than that of gastric digestion. This was due to the fact that PSPAs are transformed into more unstable quinone base and chalcone structures in the intestinal environment [50]. The results were consistent with previous studies, which showed that chalcone glycosides or chalcone are formed under pH and temperature conditions in the intestine, leading to the conversion of degradation products [9]. In the gastric digestion stage, the stability of PSPAs was high, and the effect of in vitro simulated digestion on anthocyanins was small. In the intestinal digestion stage, the structure of PSPAs transformed into quinone base, which promoted the covalent binding between PSPAs and WPHs, thus improving the stability of PSPAs in in vitro simulated digestion.

## 4. Conclusions

The present study showed that the addition of WPHs significantly improved the stability of thermal processing, storage for 7 days at 37 °C and 45 °C and in vitro digestion of PSPAs at a pH of 6.8. The addition of WPHs to composite cores improved the stability of PSPAs, confirmed by the results of the UV and fluorescence spectra, zeta potential, AFM and multiple WPH molecules. Correspondingly, the thermal degradation of PSPAs with and without WPHs followed the first-order kinetics, and the addition of WPHs effectively narrowed the degradation rate constant by 39.4 and the half-life was correspondingly extended by 1.65 times after sterilization at 121 °C. The results are important for expanding the application of PSPAs in milk products at pHs of 6.8; the stability and flavor of milk with PSPAs will also be explored in the future.

## Figures and Tables

**Figure 1 foods-13-00843-f001:**
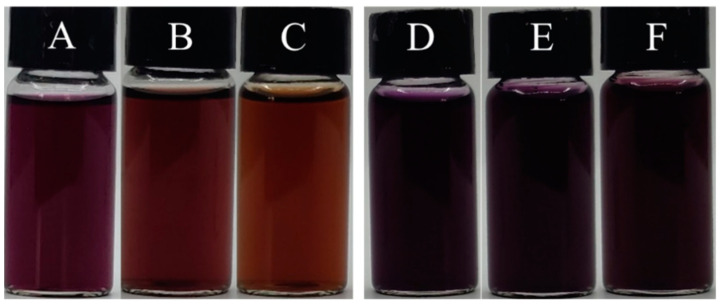
Photos of anthocyanin solutions after commercial sterilization and storage. (**A**–**C**) for PSPAs; (**D**–**F**) for WPHs+PSPAs; (**A**,**D**) after commercial sterilization; (**B**,**E**) after storage at 37 °C; (**C**,**F**) after storage at 45 °C.

**Figure 2 foods-13-00843-f002:**
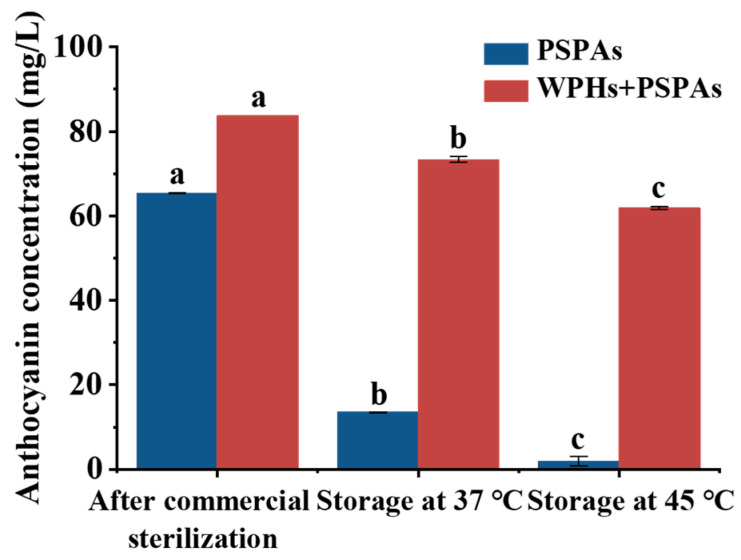
Anthocyanin concentration of solutions after commercial sterilization and storage. These numbers are the mean ± standard deviation (±SD) from the results of triplicate tests (*n* = 3). Different letters in the same set of histograms represent significant differences (*p* < 0.05).

**Figure 3 foods-13-00843-f003:**
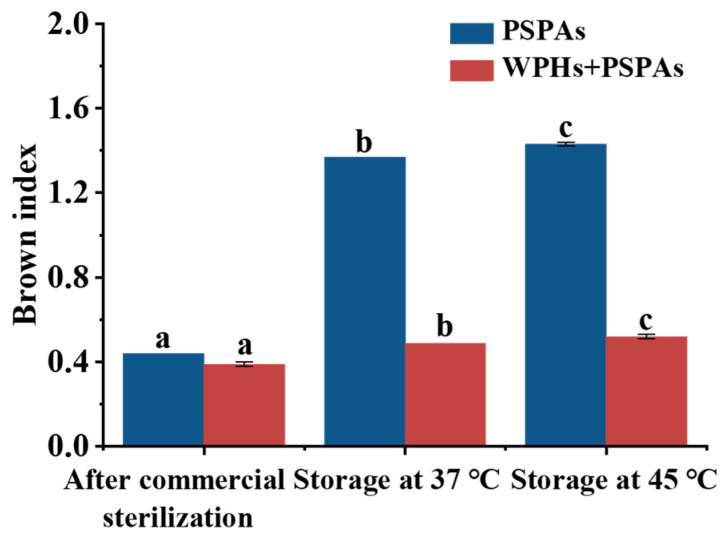
BI of anthocyanin solution after commercial sterilization and storage. These numbers are the mean ± standard deviation (±SD) from the results of triplicate tests (*n* = 3). Different letters in the same set of histograms represent significant differences *(p* < 0.05).

**Figure 4 foods-13-00843-f004:**
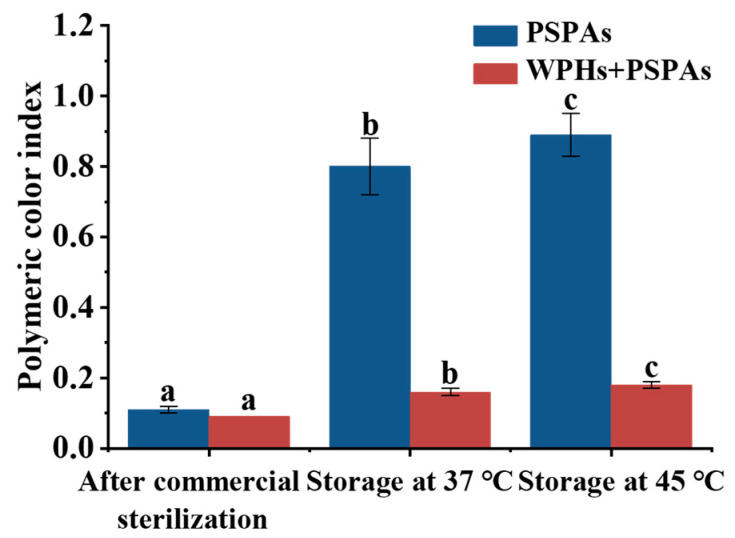
PCI of anthocyanin solution after commercial sterilization and storage. These numbers are the mean ± standard deviation (±SD) from the results of triplicate tests (*n* = 3). Different letters in the same set of histograms represent significant differences (*p* < 0.05).

**Figure 5 foods-13-00843-f005:**
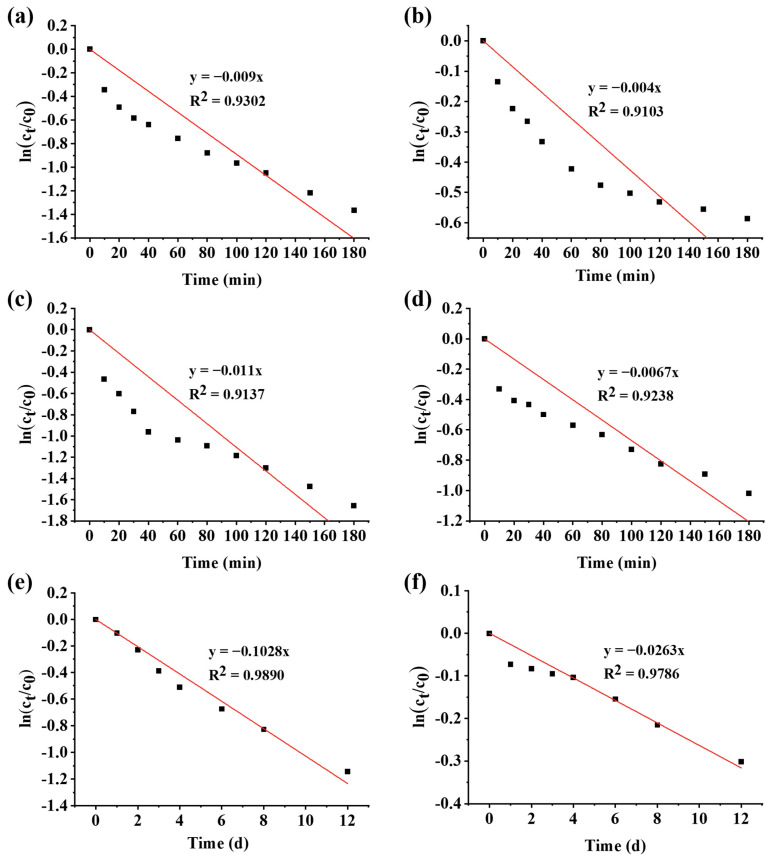
Degradation kinetics of anthocyanins. (**a**,**b**) for thermal degradation at 80 °C; (**c**,**d**) for the thermal degradation at 121 °C; (**e**,**f**) for the photodegradation at 25 °C. (**a**,**c**,**e**) for PSPAs; (**b**,**d**,**f**) for WPHs+PSPAs.

**Figure 6 foods-13-00843-f006:**
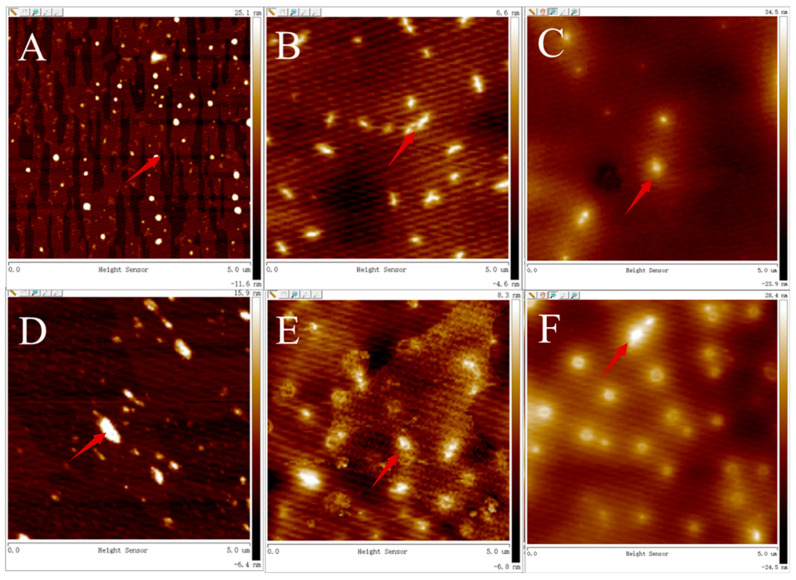
AFM topography images of PSPA solution. (**A**–**C**) for PSPAs; (**D**–**F**) for WPHs+PSPAs; (**A**,**D**) after commercial sterilization; (**B**,**E**) after storage at 37 °C; (**C**,**F**) after storage at 45 °C. The red arrows in the figure indicate PSPAs alone and PSPAs with WPHs after accelerated storage.

**Figure 7 foods-13-00843-f007:**
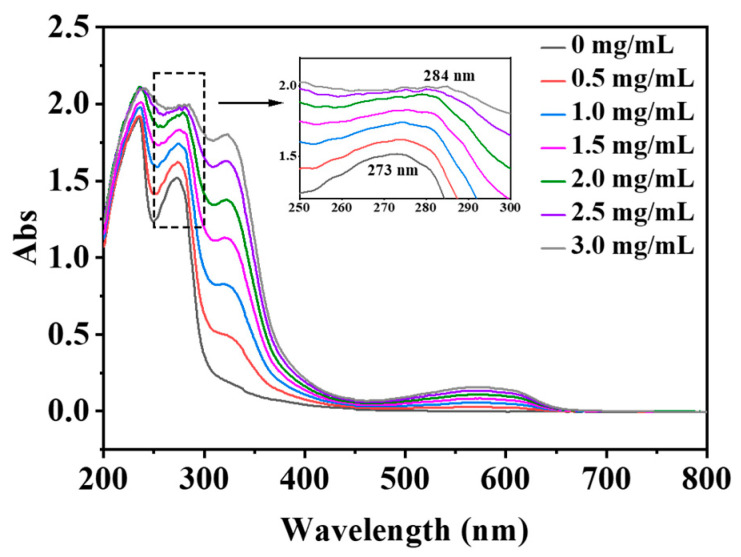
The effects of PSPAs on the UV absorption spectra of WPHs.

**Figure 8 foods-13-00843-f008:**
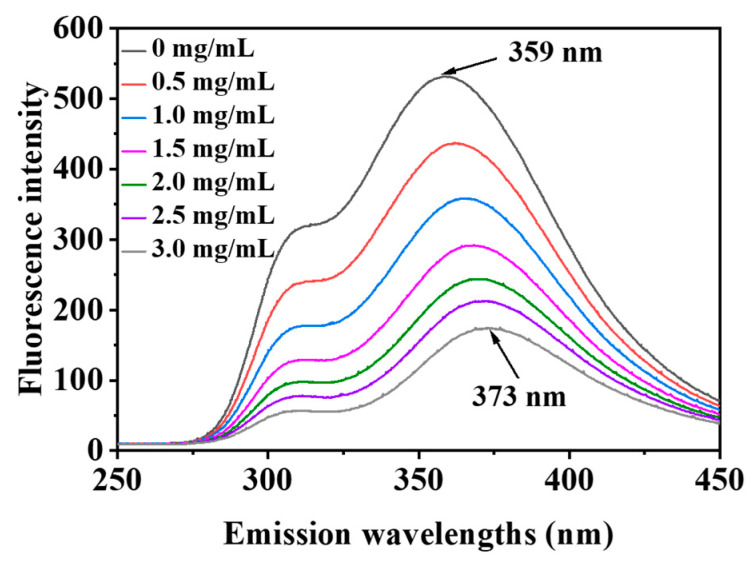
The effects of PSPAs on the fluorescence emission spectra of WPHs.

**Figure 9 foods-13-00843-f009:**
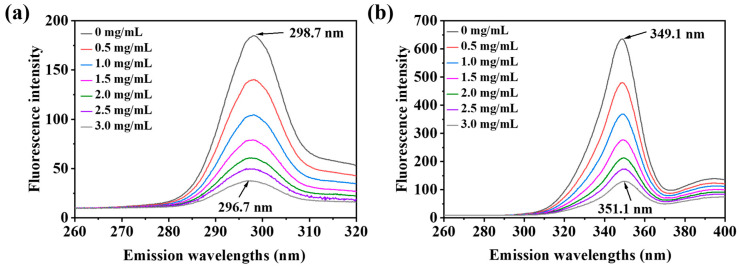
The effects of PSPAs on the synchronous fluorescence spectra at Δλ = 15 nm (**a**) and Δλ = 60 nm (**b**) of WPHs.

**Figure 10 foods-13-00843-f010:**
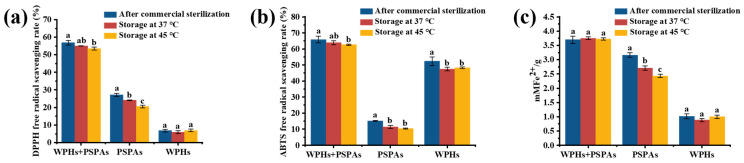
DPPH free radical scavenging capacity (**a**), ABTS free radical scavenging capacity (**b**), ferric reducing antioxidant power (**c**) of anthocyanins solutions after commercial sterilization and storage. These numbers are the mean ± standard deviation (±SD) from the results of triplicate tests (*n* = 3). Different letters in the same set of histograms represent significant differences (*p* < 0.05).

**Figure 11 foods-13-00843-f011:**
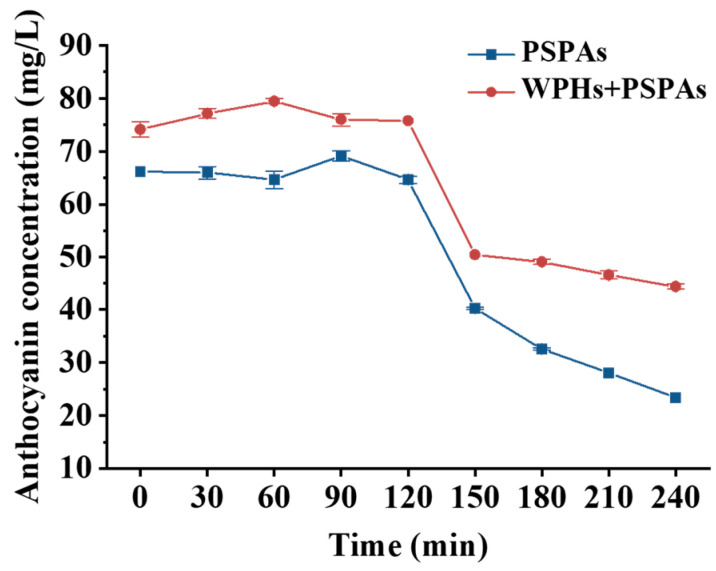
Changes of anthocyanin content in anthocyanins solution during in vitro simulated digestion.

**Table 1 foods-13-00843-t001:** Color changes of anthocyanin solutions after commercial sterilization and storage.

	PSPAs			WPHs+PSPAs		
	After Commercial Sterilization	Storage at 37 °C	Storage at 45 °C	After Commercial Sterilization	Storage at 37 °C	Storage at 45 °C
L*	26.16 ± 0.06 ^c^	28.8 ± 0.04 ^b^	30.83 ± 0.17 ^a^	25.29 ± 0.05 ^e^	25.47 ± 0.03 ^de^	25.57 ± 0.35 ^d^
a*	9.78 ± 0.45 ^c^	12.52 ± 0.13 ^b^	12.95 ± 0.25 ^a^	5.23 ± 0.12 ^d^	5.07 ± 0.17 ^d^	5.15 ± 0.25 ^d^
b*	0.59 ± 0.02 ^c^	5.46 ± 0.11 ^b^	9.38 ± 0.40 ^a^	−0.24 ± 0.05 ^d^	−0.22 ± 0.07 ^d^	0.31 ± 0.22 ^c^
ΔE	^--^	6.19 ± 0.29 ^b^	10.46 ± 0.54 ^a^	^--^	0.29 ± 0.09 ^c^	0.77 ± 0.15 ^c^
C*	9.80 ± 0.45 ^c^	13.66 ± 0.17 ^b^	15.70 ± 0.47 ^a^	5.24 ± 0.13 ^d^	5.07 ± 0.18 ^d^	5.16 ± 0.33 ^d^
h	3.45 ± 0.04 ^c^	23.56 ± 0.21 ^b^	36.69 ± 0.63 ^a^	−2.63 ± 0.61 ^d^	−2.48 ± 0.87 ^d^	3.44 ± 2.28 ^c^

These numbers are the mean ± standard deviation (±SD) from the results of triplicate tests (*n* = 3). ^a,b,c,d,e^ Different superscript letters on the same line indicate significant differences (*p* < 0.05).

**Table 2 foods-13-00843-t002:** Degradation rate constant and half-life of anthocyanins.

Sample	*k* (10^−2^ min^−1^)	Half-Life (min)
WPHs+PSPAs, sterilization at 80 °C	0.456	152.00
PSPAs, sterilization at 80 °C	0.892	77.70
WPHs+PSPAs, sterilization at 121 °C	0.671	103.30
PSPAs, sterilization at 121 °C	1.107	62.61
	***k* (10^−2^d^−1^)**	**Half-life (d)**
WPHs+PSPAs, illumination	2.634	26.35
PSPAs, illumination	10.285	6.73

**Table 3 foods-13-00843-t003:** Particle size and zeta potential of anthocyanins solutions after commercial sterilization and storage.

	PSPAs			WPHs+PSPAs		
	After Commercial Sterilization	Storage at 37 °C	Storage at 45 °C	After Commercial Sterilization	Storage at 37 °C	Storage at 45 °C
Particle size (nm)	167.58 ± 27.39 ^c^	293.68 ± 19.56 ^b^	351.21 ± 5.86 ^a^	343.13 ± 34.03 ^b^	460.91 ± 58.95 ^a^	511.89 ± 40.25 ^a^
Zeta potential	−51.01 ± 2.75 ^a^	−49.55 ± 2.34 ^a^	−48.89 ± 1.68 ^a^	−35.01 ± 2.94 ^a^	−33.49 ± 2.85 ^a^	−35.43 ± 0.06 ^a^

These numbers are the mean ± standard deviation (±SD) from the results of triplicate tests (*n* = 3). ^a,b,c^ Different superscript letters on the same line indicate significant differences (*p* < 0.05).

## Data Availability

The original contributions presented in the study are included in the article, further inquiries can be directed to the corresponding author.

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
