# Peer review of "Wheat Protein Hydrolysates Improving the Stability of Purple Sweet Potato Anthocyanins under Neutral pH after Commercial Sterilization at 121 °C"

_foods, 2024, doi:10.3390/foods13060843_

Round 1
Reviewer 1 Report
Comments and Suggestions for Authors
I send a review of manuscript ID number foods-2867092, of the authors: Yaping Feng, Bingqian Qiao, Xue Lu, Jianhui Xiao, Lili Yu and Liya Niu „Wheat protein hydrolysates improving the stability of purple sweet potato anthocyanins under neutral pH after commercial sterilization at 121 ℃”.
I think that the manuscript deals with an interesting area of scientific research on the aspect of the impact of wheat protein hydrolysates addition on stability of purple sweet potato anthocyanins under certain conditions. I think that this manuscript is relevant to the publication in Foods, but the authors should make a minor revision.
Keywords: Please add to the keywords - the word – pH
Introduction:
Page 2, lines 58-64; I ask the authors to clearly define and specify the aim of the work (now it resembles partially a introduction and partially a description of the methodology).
Materials and methods:
Page 5, lines 172-173; Please specify what statistical analysis was used to analyse the data and list what was done, e.g.: homogeneous groups were determined (e.g. see in Table 1)
- Results and disccusion:
Page 5, lines 179-181; … increased by 0.7% and 1%...; …by 10% and 17%... I think that when presenting a change in the value of the parameter "L" (colour discriminant parameter) one cannot state that the colour has increased by 0.7% or by 1% - these are not indications of e.g. the amount of chemical components whose amount has increased or decreased under the influence of some action. I think that the Authors should state how the colour of the samples changed, stating how the value of the colour discriminant "L" changed (from what, to what?). Please also address the distinguishing factors of the parameter "a" and "b".. I ask the Authors to improve this part of the manuscript.
Page 5, lines 190-192; Please add abbreviation under Table 1 regarding - standard deviation (± SD) and number of repetitions (n=…). Please please remove the lower case a before the sentence - … These numbers are… and include indication a,b,c,d,e before the sentence - … Different superscript letters… Please also apply under Table 3.
Page 8, lines 249-250; …the thermal degradation at 80 ℃…, …for the photodegradation at 25 ℃…- Have these process conditions been included in the methodological part of the manuscript?
Please add information under Figures 2, 3, 4 and 10 regarding - standard deviation (± SD) and number of repetitions – (n=?).
References
All literature is cited in the text.
Author Response
Reviewer #1: I think that the manuscript deals with an interesting area of scientific research on the aspect of the impact of wheat protein hydrolysates addition on stability of purple sweet potato anthocyanins under certain conditions. I think that this manuscript is relevant to the publication in Foods, but the authors should make a minor revision. Comment 1: Keywords: Please add to the keywords - the word – pH Response: The word – pH has been added and marked red to the keywords. Thanks. Comment 2: Page 2, lines 58-64; I ask the authors to clearly define and specify the aim of the work (now it resembles partially a introduction and partially a description of the methodology). Response: Thanks for your comment. This part has been rewritten and marked red in the manuscript (Line 59-63). Comment 3: Page 5, lines 172-173; Please specify what statistical analysis was used to analyse the data and list what was done, e.g.: homogeneous groups were determined (e.g. see in Table 1) Response: Thanks for your comment. Specifics have been added and marked red in the manuscript (Line 176-178). Comment 4: Page 5, lines 179-181; … increased by 0.7% and 1%...; …by 10% and 17%... I think that when presenting a change in the value of the parameter "L" (colour discriminant parameter) one cannot state that the colour has increased by 0.7% or by 1% - these are not indications of e.g. the amount of chemical components whose amount has increased or decreased under the influence of some action. I think that the Authors should state how the colour of the samples changed, stating how the value of the colour discriminant "L" changed (from what, to what?). Please also address the distinguishing factors of the parameter "a" and "b". I ask the Authors to improve this part of the manuscript. Response: Thanks for your comment. This part has been rewritten and marked red in the manuscript (Line 183-189). Comment 5: Page 5, lines 190-192; Please add abbreviation under Table 1 regarding - standard deviation (± SD) and number of repetitions (n=…). Please please remove the lower case a before the sentence - … These numbers are… and include indication a,b,c,d,e before the sentence - … Different superscript letters… Please also apply under Table 3. Response: Thanks for your comment. Corrected and marked red in the manuscript. Comment 6: Page 8, lines 249-250; …the thermal degradation at 80 ℃…, …for the photodegradation at 25 ℃…- Have these process conditions been included in the methodological part of the manuscript? Response: Thanks for your comment. Specifics have been added and marked red in the manuscript (Line 116-118). Comment7: Please add information under Figures 2, 3, 4 and 10 regarding - standard deviation (± SD) and number of repetitions – (n=?). Response: Thanks for your comment. They have been added and marked red in the manuscript.
Reviewer 2 Report
Comments and Suggestions for Authors
The experiment was planned very carefully. The Introduction section includes all necessary information about examined objects and problems. At the end of the introduction, the section's main goal was presented. All figures and tables are presented in a clear way. General opinion: The presented manuscript is very valuable and needs revision. Besides, the English should be improved. My comments are attached to the pdf file.

Comments on the Quality of English Language
Major revision
Author Response
Reviewer #2: The experiment was planned very carefully. The Introduction section includes all necessary information about examined objects and problems. At the end of the introduction, the section's main goal was presented. All figures and tables are presented in a clear way. General opinion: The presented manuscript is very valuable and needs revision. Besides, the English should be improved. My comments are attached to the pdf file. Response: We really appreciate for your review and acknowledgement of this paper, we have revised the manuscript in accordance with your comments and marked red in the manuscript, once again, thank you very much!
Round 2
Reviewer 2 Report
Comments and Suggestions for Authors
Accept
Comments on the Quality of English Language
Minor edit